# Plug Regime Flow Velocity Measurement Problem Based on Correlability Notion and Twin Plane Electrical Capacitance Tomography: Use Case

**DOI:** 10.3390/s21062189

**Published:** 2021-03-21

**Authors:** Volodymyr Mosorov, Grzegorz Rybak, Dominik Sankowski

**Affiliations:** Institute of Applied Computer Science, Lodz University of Technology, 90-537 Łódź, Poland; grzegorz.rybak@p.lodz.pl (G.R.); dsan@kis.p.lodz.pl (D.S.)

**Keywords:** electrical capacitance tomography, pattern, correlability, multiphase flow

## Abstract

In this paper, the authors present the flow velocity measurement based on twin plane sensor electrical capacitance tomography and the cross-correlation method. It is shown that such a technique has a significant restriction for its use, particularly for the plug regime of a flow. The major issue is with the irregular regime of the flow when portions of propagated material appear in different time moments. Thus, the requirement of correlability of analyzed input signal patterns should be met. Therefore, the checking of the correlability should be considered by such a technique. The article presents a study of the efficiency of the original algorithm of automatic extraction of the suitable signal patterns which has been recently proposed, to calculate flow velocity. The obtained results allow for choosing in practice the required parameters of the algorithm to correct the extraction of signal patterns in a proper and accurate way. Various examples of the application of the discussed algorithm were presented, along with the analysis of the influence of the parameters used on the quality of plugs identification and determination of material flow.

## 1. Introduction

The flow velocity measurements are often applied to many industrial processes, such as food, mineral, chemical, and pharmaceutical production [1,2,3,4,5]. For one phase flow, velocity measurement is a relatively simple task. However, it is still challenged in multiphase flow applications when the velocity measurement of the chosen phase is required. One of the concrete applications is plastic bottle production, where a pneumatic transport system transports granular material. One of the basic efficiency criteria for such a transport system is the maintenance of low air pressure during material propagation. Thus, continuous measurement of solid material velocity is necessary for estimating the transportation efficiency of such a system. Nowadays, in practice, different techniques for the flow velocity measurement are used, although they are still not perfect. Sometimes, invasive techniques are used. They require sensors inserted into the pipe, and as a result, they disturb the material propagation and are exposed to abrasion over time. There are also laser methods and other optical methods, but they have a limited scope of application related to its radiation issues.

One of the techniques that are not invasive is process tomography. The electrical capacitance tomography (ECT) technique is successfully applied when flow phases are not electrically conductive. In the mentioned pneumatic transport, this situation occurs. Such a technique is relatively low-cost and safe, especially its measurement sensor part. The classical tomographic image velocimeter (TIV) includes twin sensor planes mounted around an investigated pipe/vessel/column. Each sensor consists of electrodes (typically, the number of the electrodes is 8, 12 rarely) and it is used to measure the electrical capacitance between any pairs. Based on these capacitance measurements, an image reconstruction algorithm is next utilized to reconstruct the material distribution inside the investigated volume of interest. To calculate the material velocity between the sensors’ planes, a series of reconstructed images are required.

The correctness and reliability of a particle velocity measurement of multiphase flow are still a key challenge, despite the fact that currently there are many published elaborations related to this research scope [6,7,8]. Excluding the problems concerning the time stability of the measured capacitances and calibration limitations, the main problem is how to correctly calculate flow velocity based on reconstructed images.

Although different techniques are proposed, for instance, the method based on the computation of the sensitivity matrix spatial gradient [9,10], the main method of velocity calculation is still a classical cross-correlation one [11,12,13,14,15,16]. To estimate flow velocity in the chosen pixel of the volume of interest, first, the cross-correlation function of signals representing material distribution changes within the pixel of the first and second plane of the tomographic unit is calculated. It is assumed that the global maximum of the calculated function corresponds to a transit time of material propagation between two sensor planes. Moreover, it is important to mention that the transit time is calculated for a time window. Therefore, this calculated velocity corresponds to the average material velocity between two sensor planes.

The signal correlation function is based on the indicated time intervals, in which the range of measurements used in the calculations is determined. The indicated time intervals constitute time windows for the measurement data. Most often, such a window is set as a constant value before performing the calculations and its scope does not change during the analysis. Unfortunately, there are no hints or information on what basis the parameters were determined [16,17,18,19,20]. On the other hand, it turns out that the window width parameter is important in the context of ensuring the effectiveness of the algorithm. Due to this fact, it is obvious that the pattern should be computed based on the signal that is processed. Warsito and Fan 2008 [21] proposed the technique of tracing flow structures but without indicating the mechanism of time window selection. Fuchs et al. [22] described a method based on determining the length of a plug, but without explaining how to determine the length of time windows for the flow velocity estimation algorithm. The problem is still omitted by the researchers, for instance, in the articles published in the last years [4,23,24].

The existing gap has recently been filled up by developing a new concept for an appropriate algorithm allowing the automatic determination of the time window (Mosorov et al. 2020) [11]. The authors propose automatically detecting signal patterns to measure the particle velocity based on the obtained series of tomographic images. However, the proposed algorithm uses arbitrary chosen parameters, and their choices have not been justified. There is no study presenting the dependence between the accuracy of flow velocity measurement and parameter values. Therefore, the article has several aims:Justification of the choice of parameters to detect signal patterns to measure flow velocity.Determination of the problems for utilization of the proposed approach in real-case-studies.

The authors suppose that the obtained results of the conducted study can be successfully utilized in many applications that use the cross-correlation method to calculate time parameters of flow measurements, where the plug regime occurs.

## 2. Theoretical Considerations

An ECT tomography system to determine a velocity profile in the region of interest should consist of at least two plane sensors mounted around a pipe (see Figure 1). To ensure fulfilling the assumption on the laminar character of the investigated flow, a distance between the two planes should be chosen, as short as possible. Such an ECT tomographic velocimeter provides the constant speed of imagining, representing the distribution of the material concentrations in cross-sections of both planes.

Material concentrations are represented as images reconstructed based on raw measurement. The typical image resolution is 32 × 32 pixels. To calculate the flow velocity profile, the cross-correlation method is utilized. First, the cross-correlation function is calculated between changes in the corresponding pixel values of the images, and then, the maximum of the cross-correlation function is determined. Such a founded maximum determines the time delay of material propagation between the two planes.

Let $x_{i,j}\left(nT\right)$ and $y_{i,j}\left(nT\right)$ define the material distribution changes for (i,j) pixel of the nth *I* × *J* tomographic image (see Figure 1) captured with the frame rate resolution T obtained from planes X and Y of the twin plane tomograph, respectively.

As mentioned before, it is assumed that flow has a laminar character. Therefore, a strong relationship between the material distribution changes coming from the *X* plane and *Y* one exists, and the cross-correlation function $R_{i,j}\left(kT\right)$ of the two time series $\left\{x_{i,j}\left(nT\right)\right\}$ and $\left\{y_{i,j}\left(nT\right)\right\},\;$*n* = *N*, … *N* + *M* − 1 can be applied:(1)$$R_{i,j}\left(kT\right)=\frac{1}{M}\sum_{n=N}^{N+M-1}x_{i,j}\left(nT\right)y_{i,j}\left(\left(n-k\right)T\right),\;k=\ldots ,-1,\;0,\;1\ldots$$
where *M* is the number of frames determining the time window considered in the calculations.

The founded maximum of the cross-correlation *R_max_* determines the time delay $\tau_{0\;}$in a frame rate between the two signals representing material change distributions. Then, a velocity $V_{i,j}$ of the flow in (*i*, *j*) pixel of the cross-section calculated in time window [*N*, *N* + *M*] is given by the formula:(2)Vi,j¯= dτ0

It is worth nothing that it is an average velocity of material propagation in the time window, for instance, in the frame [*N*, *N* + *M*].

The changes in the concentration of the pneumatic flow material measured in the vertical pipe section have a characteristic shape, namely quasi-regular pulses similar to rectangular ones. In terms of fluid mechanics, plug flow is a basic model describing a material flowing in a pipe. In plug flow, the velocity of the fluid is assumed to be constant across any cross-section of the pipe perpendicular to the axis of the pipe. Such a plug flow model assumes that there is no boundary layer adjacent to the inner wall of the pipe. For instance, Figure 2 shows the chosen time interval of typical normalized concentration changes in the chosen pixel of the tomographic images for a pneumatic conveying, where 0 corresponds to an empty pipe and 1 means the highest material concentration (to normalize material concentration, the calibration procedure is required).

As mentioned in Mosorov et al. 2020 [11], the estimation of the time delay between two time signals requires fulfilling the condition of their correlability. The notion of correlability of input signals means that the calculated cross-correlation function will allow determining a dependency between the analyzed signals. The correlability is strictly connected with a problem determining the time window to the calculation of the cross-correlation function. This means that the choice of the time window depends on the considered signal patterns. Hence, the velocity calculation should consider choosing the appropriate signal patterns of time series $x_{i,j}\left(nT\right)$ and $y_{i,j}\left(nT\right)$, i.e., determination of a time window.

Since the plug propagation time is not constant, the choice of a time window with a fixed length *M* is not possible. In addition, plugs appear irregularly, therefore, there is a problem: How to automatically determine the time of appearance of the plug and its finishing. Hence, the algorithm for determining the velocity based on Equation (1) should automatically detect the moments of plug occurrence.

Applying simple thresholding to determine the time moments is not possible due to the fact that there can be a high probability of short pulses which are not suitable for determining the material velocity by the correlation method due to their short duration. Such a case can be described as impulsive noise in terms of signal processing notions.

In Mosorov et al. 2020 [11], the algorithm to determine the time interval is proposed. It includes the following steps:
Choose arbitrarily the threshold value *s*_0_ experimentally and an interval of confidence *m*_0_*T*, where *m*_0_ is the arbitrarily chosen number of frames. It is assumed that the threshold values must be above the known noise level.Waiting when the signal $x_{i,j}\left(nT\right)$ or $y_{i,j}\left(nT\right)$ is below the threshold value *s*_0_.If the signal value $x_{i,j}\left(nT\right)\;\mathrm{or}\;y_{i,j}\left(nT\right)\;$ exceeds the threshold value *s*_0_ during the confidence interval [*N_b_T*, *N_b_T* + *m*_0_*T*], then moment *N_b_T* − *m*_0_*T* determines the beginning of the signal pattern.If the signal value $x_{i,j}\left(nT\right)\;\mathrm{or}\;y_{i,j}\left(nT\right)$ is below the threshold value *s*_0_ during the interval of confidence [*N_e_T*, *N_e_T* + *m*_0_*T*], then moment *N_e_T* + *m*_0_*T* is chosen as the end of the signal pattern.The cross correlation function is calculated as follows:
(3)$$R_{i,j}\left(kT\right)=\frac{1}{N_{e}-N_{b}+2M_{0}}\sum_{n=N_{b}-M_{0}}^{N_{e}+M_{0}}x_{i,j}\left(nT\right)y_{i,j}\left(\left(n-k\right)T\right),\;k=\ldots ,-1,\;0,\;1\ldots \;$$Finally, the velocity of the flow in (*I*,*j*) pixel is calculated according to Equation (2), during the interval [*N_e_T*, *N_e_T* + *m*_0_*T*]:
(4)Vi,j|[NeT, NeT+M0T]¯= dτ0Return to step 2 where *n* = *N_e_*, i.e., starting from the moment of the interval when the signal $x_{i,j}\left(N_{e}T\right)\;\mathrm{or}\;y_{i,j}\left(N_{e}T\right)$ is below the threshold value *s*_0_.

In practice, it is unnecessary to combine two signal values $x_{i,j}$ or $y_{i,j}$ and it has to choose one of them. In the article, such a basic signal is $x_{i,j}$. However, there are no studies showing how the rightness of the velocity calculation will depend on the choice of *s*_0_ and *m*_0_ parameters. Therefore, the next section describes the carried out studies regarding the estimation of influence of these parameters on the rightness of velocity calculation and finding the best values for the calculation procedure.

## 3. Experiments and Discussion

The measurement is made with an electrical capacitance tomography unit by measuring the value of the electric capacity between successive pairs of electrodes *N_el_*. Electrodes are placed around the pipe at an equal distance, as shown in Figure 3. The measurement output has the form of the vector where its elements stand for consecutive electrode pairs:$$V\;=\;\left[\;\begin{array}{ccc} 1.0073 & \ldots & 1.0114 \end{array}\right].$$

The number of electrode pairs *N_el_* can be calculated as follows:(5)$$N_{el}=\;\frac{n_{e}\left(n_{e}-1\right)}{2}$$
where *n_e_* is the number of electrodes.

In our experiment, the number of electrodes *n_e_* is 8 and hence, the number of electrode pairs *N_el_* is 28.

The process of obtaining a tomogram, the graphical representation of reconstructed tomographic data requires several stages, which include: Data acquisition from tomographic sensors, device calibration (determining the range of the minimum and maximum value for the process, normalization of measurement data, preparation of the sensitivity matrix, and reconstruction of the tomographic image).

One of the ECT data processing stages is to perform an image reconstruction algorithm that allows preparing the pipe cross-section view. There are many such methods. However, despite the lower quality, the most frequently used method in real-time systems is the linear back projection (LBP) method [25,26] due to the low cost of computing. The method of linear back projection is a method belonging to the group of inverse problem solution [27], allowing to calculate more result data than the amount of input data. This method includes calculating all points of the tomogram using the so-called sensitivity matrix. This structure is based on a predefined N-element image table, where each element corresponds to the distribution of electric capacitance between two sensor electrodes. Part of the sensitivity matrix with example pairs of electrodes are shown in Figure 4.

The sensitivity matrix is a graphical representation of the electric capacitance distribution between the electrodes, which results in the amplification effect of the values between the electrodes, while weakening the signal’s influence on the central area of the analyzed plane. The size of each sensitivity matrix image is equal to 32 × 32 pixels of the same dimensions, as a result, a reconstructed image. The output is obtained thanks to Equation (6):(6)$$IMG_{Mx1}=S_{L\times M}^{-1}*C_{Lx1}$$
where *M* is the ECT image resolution, *L* is the vector size of one ECT measurement, *C_Lx_*_1_ is the ECT measurement vector, *S_L×M_* is the sensitivity matrix, and *IMG* is the tomographic image.

There are many numerical problems associated with the sensitivity matrix. This matrix is not a square matrix. Its dimensions depend on the number of independent measurements and from a significantly greater number of image points. Hence, it is impossible to calculate the inverse of matrices and in various known methods the pseudo-reverse of the image reconstruction is used. The problem is considered as ill-conditioned. In the LBP algorithm, the pseudo-inverse sensitivity matrix is computed through its transpositions.

After performing the image reconstruction algorithm, the cross-section image is obtained (Figure 5a).

Thanks to ECT, using the LBP method, it is possible to determine the number of substances in the transport route or the distribution of material concentration in the cross-section of the tank during industrial processes. Despite the low quality, the accuracy of concentration changes is determined at the level of about 3%.

The research was based on a two-phase pneumatic gas-bulk material flow. The equipment configuration includes two 100 L tanks placed one below the other, a rotary feeder, a pump, piping in the horizontal section (2 × 7 m^2^) and vertical (approx. 3 m), as shown in Figure 6. Stainless steel piping was used with a diameter of Ø1 = 57 mm (inner). In the upper part of the installation, there is a filter that separates the phase of the loose material from the gaseous medium. A polymer granulate with a size of 3 mm × 3 mm × 1 mm and a dielectric permittivity of approximately 2.1 was adopted as the transported substance (Jaworski and Dyakowski 2001) [28].

As seen in Figure 6, the tomographic sensors have been installed in the vertical section. Each of them consists of eight electrodes placed at equal intervals on the outer wall (electrode width 30 mm). The distance between them is constant and was *d* = 130 mm. The measurement was made using the mentioned tomographic unit with a sampling frequency of 100 Hz. The gas velocity (for an empty pipe) can vary from 1.0 to 5 m/s by inserting three different nozzles and the pressure control. It follows from the above that the highest achievable material velocity is 5 m/s.

As part of the experiment, the acquisition of measurement values was performed using a tomographic sensor from two planes, and then the data were processed according to the steps shown in Figure 1. The main parts of the algorithm are the plugs identification section and the cross-correlation module. When the measurement samples arrive at the plug identification module, the *s*_0_ threshold is used to determine if the plug appears. Then, if the higher material concentration is recorded in the period greater than *m*_0_ (confidence threshold), the plug beginning is indicated. The opposite procedure is performed for plug end determination. Figure 7 presents example graphs showing changes in the concentration value in the pipe cross-section based on a 15 × 15 pixel of tomographic images from two planes and the result of the discussed plug identification algorithm to parameters *s*_0_, *m*_0_. As can be seen, the pattern of the time range (dotted red line) covers the time range of plug appearance (blue line) completely.

The discussed algorithm, together with arbitrarily set parameters, apparently indicates the correct speed estimates. However, there are situations when the obtained values are inconsistent with reality. One of the reasons for the occurrence of inappropriate estimates is the reliance on incorrect algorithm arguments. The analysis of the plug registration threshold *s*_0_ impact on the plugs identification algorithm effectiveness was performed. The results are presented in Figure 8a,b.

Using the parameters *s*_0_ = 0.35 and *m*_0_ = 5, an additional wrong plug is detected (Figure 8a). It results from the occurrence of situations where the instantaneous fluctuations of the concentration level exceed the indicated threshold and remain above *m*_0_ = 5 units. In such a situation, a signal pattern is generated, where the maximum correlation reaches the value of *n* = 21 (wrong velocity value eq. 0.62 m/s). The plug identification algorithm requires proper parameter setting that would enable the correct speed estimation. The authors conducted a series of experiments that allow indicating the right values. The results are shown in Figure 9.

The conducted experiments show unequivocally that the use of the right algorithm parameters is crucial when assessing its effectiveness. For the real data series, which shows the passage of 20 plugs (Figure 9a red line), it is noticeable that for the threshold *s*_0_ ≥ 0.75 the number of identified plugs does not reach the expected value and additionally shows an extreme influence of the *m*_0_ parameter. Moreover, for a value of *s*_0_ ≤ 1, a similar situation exists.

On the other hand, if there is a higher *s*_0_ threshold, then a lower *m*_0_ parameter should be set to return the required number of plugs. Additionally, reducing the *m*_0_ allows identifying more plugs.

The second experiment was performed in order to visually estimate the signal time shifts (*p*) for successive plugs. Then, results were taken as reference values to calculate the root of mean square error (RMSE); Equation (7)) for the algorithm with respect to *s*_0_, *m*_0_.
(7)$$RMSE=\sqrt{\frac{\sum_{n=1}^{Pn}{\left(\hat{p}-p\left(n\right)\right)}^{2}}{Pn}},$$
where *P_n_* is the plug count; *n* is the plug number; p(n) is the time shift for *n*-th plug between planes; $\hat{p}$ is the expected time shift (visual analysis).

As a result of the analysis, the diagram (see Figure 9b) was obtained. The reference data (visual analysis) may be inaccurate. Therefore, the values on the graph fluctuate around *RMSE* = 1. It is noticeable that for low *m*_0_ and high *s*_0_ values, the estimation error is lower. Based on the above analyses, it can be assumed that the best parameters of the discussed algorithm are *s*_0_ = 0.6 and *m*_0_ = 10.

Another aspect of the algorithm’s efficiency is the speed estimation error based on the correlation result resolution. The maximum speed that can be obtained with the indicated measurement parameters of the system equals *n*_1_ = 13 m/s (where *n* = 1 indicates the number of units of shifting the signals between two measurement planes). The following values are: *n*_1_ = 13 m/s, *n*_2_ = 6.5 m/s, *n*_3_ = 4.3 m/s, *n*_4_ = 3.25 m/s, *n*_5_ = 2.6 m/s, *n*_6_ = 2.17 m/s, *n*_7_ = 1.86 m/s, etc. Hence, the speed estimation error resulting from the erroneous calculation of the maximum value of the correlation increases with the increase in speed, reaching the maximum absolute value equal to *v* > 13 m/s (100%). In the current experiment, where the velocity of the medium is known, it is expected to obtain the plug velocity *v* < 6.5 m/s, which gives an error of 51%, and for the value of *v* = 2.17 m/s (*n* = 6), respectively 17%.

The authors conducted an algorithm accuracy analysis. The first step was to check the changes in the estimated velocity to the parameter *s*_0_. It was expected to show that different velocity estimates were obtained for different threshold values. In the second step, an error resulting from the estimation using of the considered algorithm to visual measurements was demonstrated.

The results of the experiments are presented in Figure 10. The top chart (a) shows the relative error from the elaborated algorithm.

The analysis proves that for a low threshold (smaller than 0.4) and higher than 0.7 results are not stable. Contrary, the center values give constant velocity estimation that allows us to assume this range as correct. On the lower plot, the error rate was calculated based on the reference measurement performed as a visual analysis. Same as above, a threshold greater than 0.4 gives better results. For the discussed case, the average error oscillates around 13.3% and the maximum error reaches up to 33.3%.

## 4. Conclusions

Although an ECT tomograph system can be successfully utilized to determine a velocity profile in multiphase flow, however, there still is a lack of the efficient algorithms implemented in such a system to calculate the velocity profile in the right way. It is shown that the key factor is the choice of appropriate time intervals for velocity calculations. This carried out research concerns the study of dependence between the threshold level parameters of signal pattern detection and rightness of the calculated velocity values. It is shown that these threshold level parameters have a major influence on the appropriate plug detection, as well as on the cross-correlation result that stands for accurate velocity estimation. In this article, a use case is presented which proves the relation between the *s*_0_ threshold and the *m*_0_ parameter given in the elaborated algorithm. The larger *s*_0_, then smaller *m*_0_ must be to obtain a valid count of plugs. Additionally, the correlation output is also determined by a threshold value. For the range of *s*_0_ = <0.4, 0.7>, the algorithm gives the lowest error referring to the visual analysis.

Although the conducted study was obtained for gravity flow experiments and pneumatic conveying, other studies can also be valuable for flows with different tomographic modalities.

## Figures and Tables

**Figure 1 sensors-21-02189-f001:**
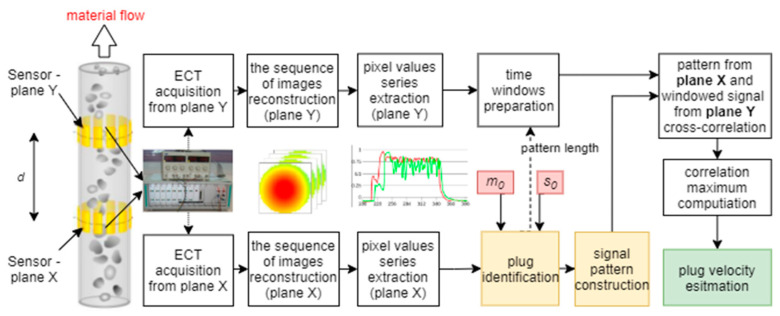
Diagram of flow velocity measurement realized by a two-plane-sensor electrical capacitance tomography (ECT) tomography, *d* is the distance between the two planes.

**Figure 2 sensors-21-02189-f002:**
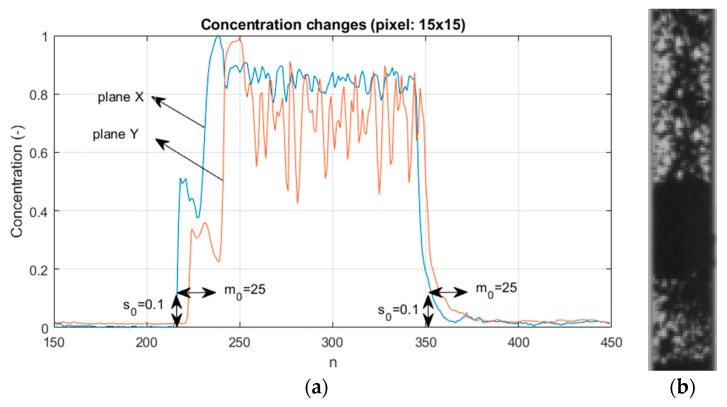
Normalized concentration changes within a chosen pixel (15,15) of the tomographic images (**a**) and the image captured by a high speed camera (#220–#320) (**b**); *s*_0_—some threshold level and *m*_0_—the arbitrarily chosen number of frames.

**Figure 3 sensors-21-02189-f003:**
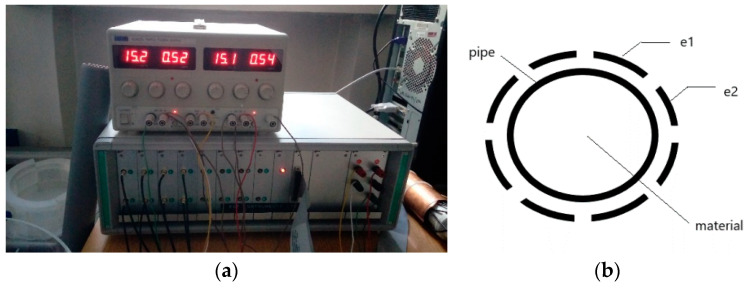
Electrical capacitance tomography measurement unit and sensor placing on the pipe (**a**). On the right hand, the cross-section of the pipe is presented with electrodes (e1, e2 …) mounted around the pipe (**b**).

**Figure 4 sensors-21-02189-f004:**
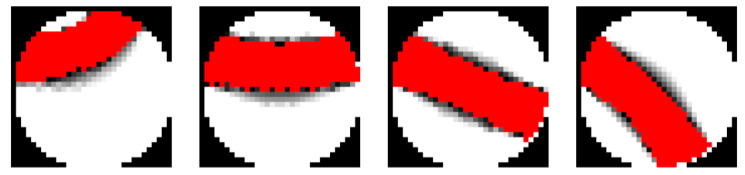
Visualization of sensitivity matrix parts, an example. The red color stands for amplified values between pairs of electrodes.

**Figure 5 sensors-21-02189-f005:**
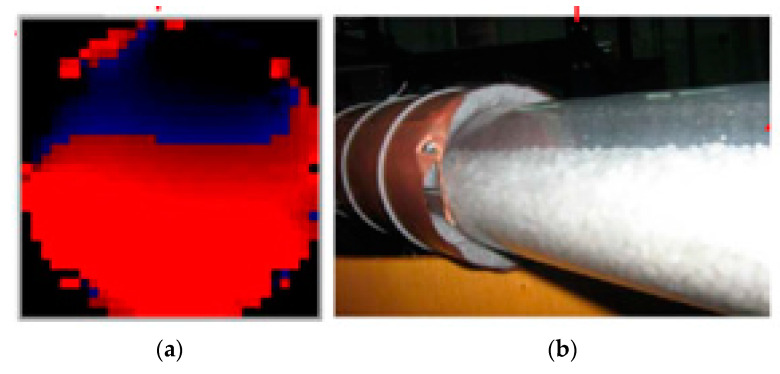
Reconstructed ECT image (**a**) and the photography that presents the material inside the pipe. (**b**) The red color shows the material inside the pipe, and the black color represents an empty part of the pipe. On the right hand, (**b**) the image of the pipe section is presented.

**Figure 6 sensors-21-02189-f006:**
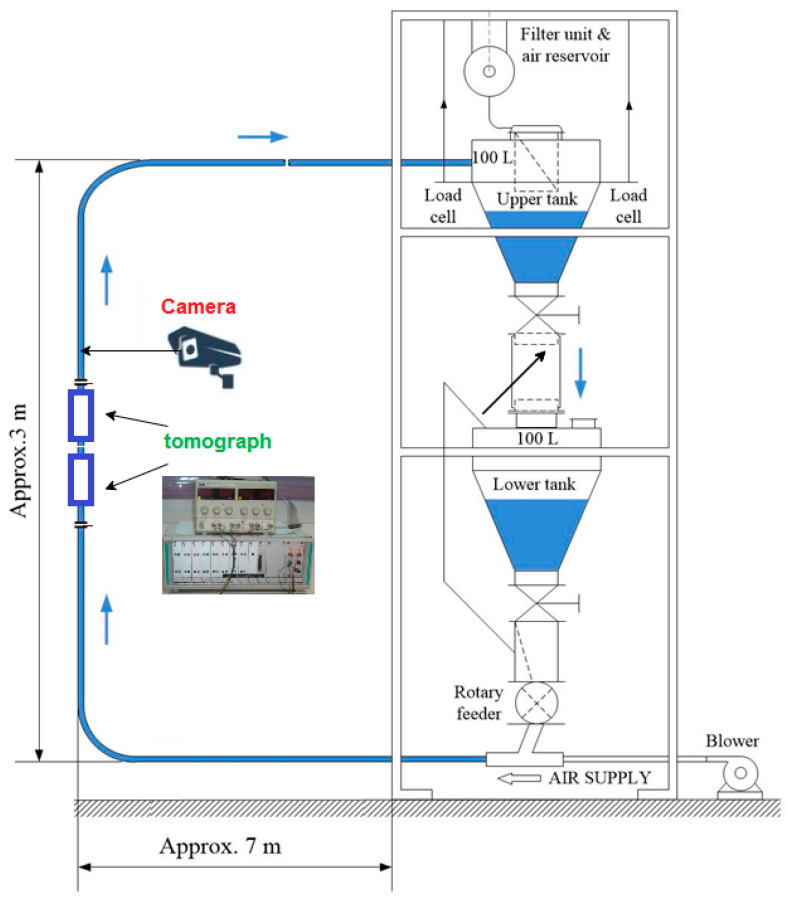
Pneumatic material conveying system schema with tomography sensors and video camera placement.

**Figure 7 sensors-21-02189-f007:**
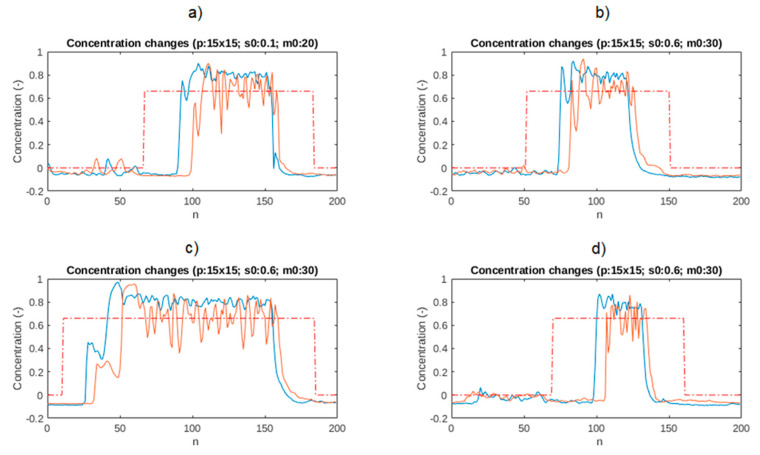
Plug identification algorithm output for several cases. The signal from the second plane is presented with an orange color. The dotted lines present ranges of computed patterns: (**a**) Shows patterns where *s*_0_ = 0.1 and *m*_0_ = 20; (**b**–**d**) shows patterns with *s*_0_ = 0.6 and *m*_0_ = 30.

**Figure 8 sensors-21-02189-f008:**
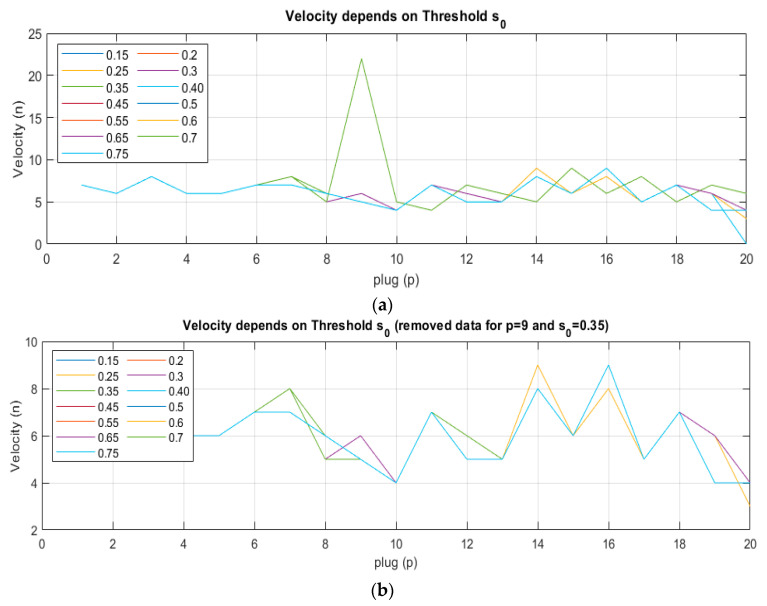
Material velocity estimation chart in relation to the parameter s_0_ for each plug. The different colors represent parameter *s*_0_: (**a**) Full data, (**b**) removed data for *s*_0_ = 0.35 m, and plug number 9.

**Figure 9 sensors-21-02189-f009:**
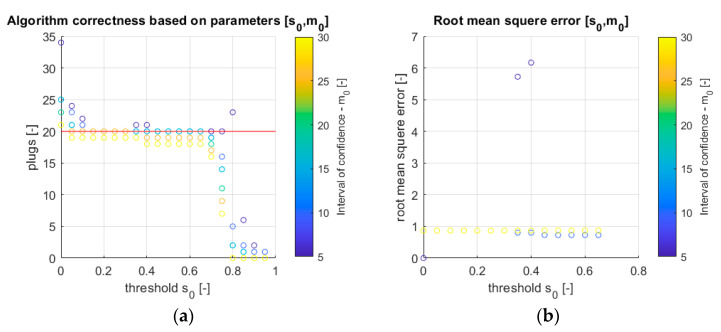
(**a**) Analysis of the influence of parameters s_0_, *m*_0_ on the number of identifiable plugs (red line—expected value); (**b**) absolute error for the correlation function with respect to parameters *s*_0_, *m*_0_. The color of circles indicates the value of m_0_ parameter (the scale is shown on the right hand).

**Figure 10 sensors-21-02189-f010:**
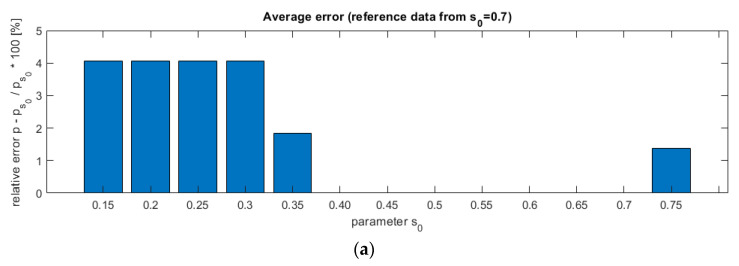
Plug identification algorithm threshold value impact on the velocity estimation accuracy. The plot on the top (**a**) shows the relative error for *s*_0_ = 0.7 as a reference (expected value). The relative error based on the visual analysis as a reference is shown beneath (**b**).

## Data Availability

Not applicable.

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
