# Peer review of "Plug Regime Flow Velocity Measurement Problem Based on Correlability Notion and Twin Plane Electrical Capacitance Tomography: Use Case"

_sensors, 2021, doi:10.3390/s21062189_

Round 1

Reviewer 1 Report

The topic of the article is plug flow velocity measurement with dual plane ECT and cross-correlation. The authors present an analysis of errors in dependence on threshold values. Certainly, this is an interesting and relevant topic. However, the paper cannot be published this way.

It is clear, that for the proposed algorithm (detection of plugs via threshold before setting a time window for x-corr) some inappropriate threshold values may create errors. The extent of this problem depends on the appearance (shape) of the plugs in the data. This holds for any detection technique and is not specific for ECT.

The authors use a single set of data from ECT to show this problem. Neither a deep analysis nor an improved aproach is presented. This is too little!

English language is not sufficient.

In many places writing is ambiguous or unclear, not only lingual. Examples are:

What means "the signal value x()/y()" on page 4 and 5? Is it a division? I would rather interpret it as x() or y().

A laminar flow has a parabolic profile, not a plug one!

Why is it a pixel-wise correlation, why not correlating cross-sectional averages?

Physical quantities need to be clarified! What is the data representing, a normalized density, a normalized permittivity, ...?

Why "concentration changes". Why concenration at all. What is concentrated?

What means that s_0 and m_0 are arbitrarily chosen? Why not basing them on some assumptions at least?

Why does N_bT+m_0T determine the beginning of signal pattern (p5, top). What means signal pattern?

....

I am sory but I cannt recommend the manuscript for publication.

Author Response

# 

Remark 

Answer 

1 

It is clear, that for the proposed algorithm (detection of plugs via threshold before setting a time window for x-corr) some inappropriate threshold values may create errors. The extent of this problem depends on the appearance (shape) of the plugs in the data. This holds for any detection technique and is not specific for ECT. 

Yes, we agree. As mentioned in your remark we cite "the extent of this problem depends on the appearance (shape) of the plugs in the data". Therefore, the study of the cited above problem i.e., the influence of inappropriate threshold values, is a goal of the article.   

2 

The authors use a single set of data from ECT to show this problem. Neither a deep analysis nor an improved aproach is presented. This is too little! 

The authors have a huge file recording the half of hour experiment. The investigated flow had the same phenomena therefore, in our opinion, the chosen set of data represent the pneumatic conveying flow enough. 

3 

In many places writing is ambiguous or unclear, not only lingual. Examples are: 

What means "the signal value x()/y()" on page 4 and 5? Is it a division? I would rather interpret it as x() or y(). 

The ambiguity has been deleted and an explanation was inserted (see p. 5) 

4 

A laminar flow has a parabolic profile, not a plug one! 

Because the velocity is measured in each image pixel, therefore, the flow profile was not considered in the article. The applying introduced notion of "correlability" allows excluding the calculation of velocity in cases when signals are not suitable. 

5 

Why is it a pixel-wise correlation, why not correlating cross-sectional averages? 

The flow velocity values are not the same in the pipe cross-section. Moreover, tomography tools allow a velocity profile calculation in the volume of interest instead of average values. 

6 

What means that s_0 and m_0 are arbitrarily chosen? Why not basing them on some assumptions at least? 

Yes, agreed, sometimes some assumptions are default. We also assume that normalized threshold values should be higher than the noise level. An additional explanation was added.  

7 

Why does N_bT+m_0T determine the beginning of signal pattern (p5, top). What means signal pattern? 

It is not correct. In text we cite  “...moment NbT-m0T determines beginning of signal pattern” i.e., not +”  but -. This assumption means that the considered signal will have a pulse-shape form. 

Many publications assume that input data can automatically be used for further calculations. In our opinion, it is not true.  

Therefore, it was introduced the notion of "correlibility" of input signals. In terms of the article, the signal pattern means that it is suitable for cross-correlation function calculation.  

Reviewer 2 Report

Comment

  The paper presents the flow velocity measurement based on twin plane sensor electrical capacitance tomography and cross-correlation method, and the obtained results allow for choosing in practice the required parameters of the algorithm to correct the extraction of signal patterns in a proper and accurate way. This research is interesting for people working on ECT technique. However, consider the following issues,

  • At present, it seems to be popular to calculate the separated phase velocity of multiphase flow based on cross correlation principle is no matter what flow pattern. The author only studies the plug flow and assumes the laminar character of the investigated flow, In fact, the above conditions may not be satisfied, as long as the multiphase flow is assumed to be stable, i.e. the so-called 'fixed flow', for example, it is widely used in the monitoring of vulcanization process in fluidized bed, and Its flow pattern is not necessarily plug flow pattern.
  • The key problem of this research is the efficiency of flow pattern identification, so the accuracy of the data obtained by the sensor and the resolution of reconstruction algorithm is very important, a simple LBP algorithm may not be appropriate.
  • The meaning of formula (7) is not clear, and the different color marks in the figure 9 should be explained.
  • Is this study applicable to actual gas-solid flow measurements, such as vulcanization processes, how to correct measurement results?
  • There are still some errors in the paper which should be corrected,

This is expectant that the manuscript would be further improved.

Author Response

# 

Remark 

Answer 

1 

At present, it seems to be popular to calculate the separated phase velocity of multiphase flow based on cross correlation principle is no matter what flow pattern. The author only studies the plug flow and assumes the laminar character of the investigated flow, In fact, the above conditions may not be satisfied, as long as the multiphase flow is assumed to be stable, i.e. the so-called 'fixed flow', for example, it is widely used in the monitoring of vulcanization process in fluidized bed, and Its flow pattern is not necessarily plug flow pattern 

The article pays attention to plug flow only. It is underlined in the title. 

To our knowledge, the cross-correlation function can be applied to all input data; howeverpractical use of cross-correlation of input data has sense when such function has an extremum (minimum/maximum). 

Otherwise, such a cross-correlation function is not useful. Therefore, not all input data can be analyzed by the cross-correlation technique. Without previous analysis of the suitability of considered input data, as shown in the article, the cross-correlation technique will not be applicable. It is worth to underline that cross-correlation can be calculated for a chosen time window. But such a time window may not cover suitable input signals. Hence the choice of a time window that covers suitable input signals called in the article as patterns, plays a crucial role. We suppose that the choice of the appropriate time window is also considered the monitoring of the vulcanization process in a fluidized bed. Unfortunately, we do not know more details. 

2 

The key problem of this research is the efficiency of flow pattern identification, so the accuracy of the data obtained by the sensor and the resolution of reconstruction algorithm is very important, a simple LBP algorithm may not be appropriate. 

Agreed. We assume the noised signals may have an influence on measurement accuracy, but the article concerns a flow velocity measurement, and we are not sure that image quality plays a crucial role in the accuracy of velocity measurement. 

3 

The meaning of formula (7) is not clear 

Added description to formula (p(n)- time shift for n-th plug between planes) 

4 

Is this study applicable to actual gas-solid flow measurements, such as vulcanization processes, how to correct measurement results? 

The approach is applicable to use in practice when measured signals (data) are quasi-stationary or nonstationary processes. If a studied process has a stationary character, there will not be necessary to apply the proposed approach, and enough to consider a choice of the suitable time window. 

5 

different color marks in the figure 9 should be explained. 

Added to text: The colour of circles indicates the value of m0 parameter (scale shown on right hand) 

6 

There are still some errors in the paper which should be corrected, 

This is expectant that the manuscript would be further improved. 

The found errors are corrected 

Reviewer 3 Report

The manuscript provides a useful study of velocimetry measurements applied to ECT sensing of fluid flows. The results are useful and should be of  interest to other practitioners in the field. I suggest publication after the revisions below:

(1) In the caption of the figures (for example Fig. 2), the itemization (a,b) should come before their description, not after.

(2) You should indicate that the expression for the voltage vector provided just before eq. (5) is simply and example, "...for example..."

(3) The sensitivity matrix is rectangular (non-square) and strictly speaking not invertible because of that. So, please clarify the meaning of the inverse operation indicated in eq. (6). Is this a pseudo-inverse?

(4) The numbering in the list of references and in the body of the text is offset by one for many references. For example, Fusch et al. is indicated as [18] in the text but it is listed as [19] in the list of references.

(5) At the top of Fig. 8, what is "depens"? Please spell it out.

(6) On top of Fig. 9(a), please spell it out.

(7) Fig. 10: "error ratio" => "relative error". 

(8) The English language usage should be improved.

Author Response

# 

Remark 

Answer 

1 

In the caption of the figures (for example Fig. 2), the itemization (a,b) should come before their description, not after. 

The text was corrected according to the reviewer’s remark. 

2 

You should indicate that the expression for the voltage vector provided just before eq. (5) is simply and example, "...for example..." 

The text was corrected according to the reviewer’s remark 

3 

The sensitivity matrix is rectangular (non-square) and strictly speaking not invertible because of that. So, please clarify the meaning of the inverse operation indicated in eq. (6). Is this a pseudo-inverse? 

There are many numerical problems associated with the sensitivity matrix. This matrix is not a square matrix. Its dimensions depend on the number of independent measurements and from a significantly greater number of image points. Hence 

it is impossible to calculate the  

the inverse of matrices and in various known methods the pseudo-reverse of the image reconstruction is used 

The problem is considered as ill-conditioned and sub-defined. In the LBP algorithm, pseudo-inverse sensitivity matrix is computed 

through its transpositions. 

4 

The numbering in the list of references and in the body of the text is offset by one for many references. For example, Fusch et al. is indicated as [18] in the text but it is listed as [19] in the list of references. 

All list of reference was corrected 

5 

At the top of Fig. 8, what is "depens"? Please spell it out. 

The article was corrected according to the reviewer’s remark. 

6 

On top of Fig. 9(a), please spell it out. 

The article was corrected according to the reviewer’s remark. 

7 

Fig. 10: "error ratio" => "relative error". 

The article was corrected according to the reviewer’s remark. 

8 

The English language usage should be improved. 

The English have been improved 

Reviewer 4 Report

The flaw noninvasive velocity measurements are the main issue of this paper. Such measurement are very important for many branches of modern industy like chemical or farmaceutical ones. That is why in my opinion, this paper is relevant and very interesting for many readers, particularly in pandemic period (pharmaceutical industry).

This topic is original and only few research teams are working on this topic.

For the flow velocity measurement problem one of the techniques that are not invasive is process tomography based on capacitance tomography. Particularly when the medium is not conducting.

The authors propose a new concept for an appropriate algorithm allowing the automatic determination of the time window. The authors also propose automatically detecting signal patterns to measure the particle velocity based on the obtained series of tomographic images.

The paper is well written and the authors presented selected results which support their conclusions.

Author Response

Thank you for your positive review

Reviewer 5 Report

This manuscript is not very well prepared. The language is not fluent, the overall illustration is not easy to follow. The authors could not present the contents clearly. 

For example, in Fig. 8(b), it is not clear what is p=9. 

This manuscript should not be considered for publicaion.

Author Response

Remark 

Answer 

For example, in Fig. 8(b), it is not clear what is p=9. 

Right, the description could be ambiguous. Corrections made in the form of a new drawing description. 

Round 2

Reviewer 5 Report

The revised manuscript can be published.